# Synthesis of pH-Sensitive Cross-Linked Basil Seed Gum/Acrylic Acid Hydrogels by Free Radical Copolymerization Technique for Sustained Delivery of Captopril

**DOI:** 10.3390/gels8050291

**Published:** 2022-05-08

**Authors:** Shazia Akram Ghumman, Sobia Noreen, Huma Hameed, Mervat A. Elsherif, Ramla Shabbir, Mavra Rana, Kashaf Junaid, Syed Nasir Abbas Bukhari

**Affiliations:** 1College of Pharmacy, University of Sargodha, Sargodha 40100, Pakistan; huma.hameed@uos.edu.pk (H.H.); mavrarana80@gmail.com (M.R.); 2Institute of Chemistry, University of Sargodha, Sargodha 40100, Pakistan; sobia.noreen@uos.edu.pk; 3Chemistry Department, College of Science, Jouf University, Sakaka 72388, Saudi Arabia; maelsherif@ju.edu.sa; 4Faculty of Pharmacy, University of Lahore, Lahore 54760, Pakistan; ramlakashif@hotmail.com; 5Department of Clinical Laboratory Sciences, College of Applied Medical Sciences, Jouf University, Sakaka 72388, Saudi Arabia; kjunaid@ju.edu.sa; 6Department of Pharmaceutical Chemistry, College of Pharmacy, Jouf University, Sakaka, 72388, Saudi Arabia

**Keywords:** basil seed gum, captopril, hydrogel, swelling behavior, kinetics

## Abstract

The pH-sensitive polymeric matrix of basil seed gum (BSG), with two different monomers, such as acrylic acid (AA) and *N*, *N*-Methylene-bis-acrylamide (MBA), was selected to use in hydrogels preparation through a free radical copolymerization technique using potassium per sulfate (KPS) as a cross linker. BSG, AA and MBA were used in multiple ratios to investigate the polymer, monomer and initiator effects on swelling properties and release pattern of captopril. Characterization of formulated hydrogels was done by FTIR, DSC/TGA, XRD and SEM techniques to confirm the stability. The hydrogels were subjected to a variety of tests, including dynamic swelling investigations, drug loading, in vitro drug release, sol–gel analyses and rheological studies. FTIR analysis confirmed that after the polymeric reaction of BSG with the AA monomer, AA chains grafted onto the backbone of BSG. The SEM micrographs illustrated an irregular, rough, and porous form of surface. Gel content was increased by increasing the contents of polymeric gum (BSG) with monomers (AA and MBA). Acidic and basic pH effects highlighted the difference between the swelling properties with BSG and AA on increasing concentration. Kinetic modelling suggested that Korsmeyer Peppas model release pattern was followed by the drug with the non-Fickian diffusion mechanism.

## 1. Introduction

In recent decades, the pharmaceutical industry has perceived a progressive interface amongst the polymeric and material science field, which highlighted the impact of novel drug delivery systems (NDDS) development [1]. An extensive choice of NDDS (such as liposomes, exosomes, niosomes, microvesicles, nanoparticles, microcapsules or hydrogels) give rise to substantial benefits and rare restrictions [2]. However, hydrogel-based polymeric systems for drug delivery are getting attention as controlled drug delivery targets to accomplish the desired requirements of pharmaceuticals [3]. Hydrogels hold very alluring physicochemical attributes that make them reasonable for a wide scope of biomedical applications [4,5,6]. Recently, smart hydrogels that respond to physiochemical environmental stimuli, including pH, temperature, light, chemicals, magnetic and electric flux, have received a lot of interest [7]. One of most important physical/chemical stimuli employed in biotechnology and medicinal uses are pH, as well as temperature and, hence, these hydrogels were intensively explored [8]. Several synthetic pH and temperature-sensitive polymers, on the other hand, are not biodegradable, which poses a severe constraint in several applications. As a result, the creation of biodegradable pH and temperature-sensitive hydrogels derived from natural polymers has gotten a lot of attention [7,9]. A smart orally administered delivery system might be based primarily on both of these pH-dependent swelling characteristics and temperature sensitivity [10,11]. Cross-linked polymeric networks are known as hydrogels having hydrophilic groups in large numbers. These gels actually represent the higher water-imbibing capacity and with physical/chemical linkages within the polymeric chains, never allowing them to dissolve in water [12,13,14]. When water permeation happens in these networks, swelling occurs and pore formation results. This pore formation favors the drug loading into the gel matrix. Further, the release of the drug pattern from such gel matrices are dependent on the diffusion coefficient linked to micro-molecules or macro-molecules [15].

At present, polymers of natural origin are gaining more interest in drug development techniques because of their non-toxic, biocompatible and biodegradable nature, especially polysaccharides. For the manufacture of hydrogels and hydrogel-based polymers, a variety of polymeric materials, both natural and synthetic, are accessible. Hydrogels and hydrogel-based materials have shown potential competence for targeted drug delivery [16,17,18]. Natural polymers, such as alginate acid, guar gum, xanthan gum, agar and chitosan, achieved tremendous control drug delivery; these biopolymers, in combination with polyacrylamide, polymethacrylic acid, polyvinyl alcohol, polyethylene glycol and polyacrylic acid have proven to be successful in the delivery of controlled drugs [19,20].

Researchers has assured wonderful advancements via polymeric hydrogels through atom transfer radical polymerization, graft copolymerization and chemical cross linking [21]. These methods have been widely used; however, graft co-polymerization is a suitable and easy technique for the preparation of porous hydrogels [22,23]. By grafting natural polysaccharides with synthetic polymers, one can benefit from the best of both substances. Various polysaccharides from natural sources were improved by various monomers and formed potent pH-responsive hydrogels via free radical polymerization based on this concept [24]. In solution polymerization or cross-linking techniques, monomers containing free radicals exhibit network ability [25]. Providing heat/UV rays or by adding initiator, free radicals generate and initiate the process of forming a polymer matrix. Different cross-linking properties are present in both monomers and polymers or ionic or neutral nature. The solvent (such as water/ethanol or its mixture used) in polymerization serves as a passive heat exchanger, transferring the heat to fluid. After this, the formulated hydrogels are washed excessively with distilled water, so that the unreacted monomer, polymer, initiator and impurities are washed away [26,27].

Basil (*Ocimum basilicum* L.) belongs to the genus *Ocimum*. It is actually one of the common plants produced in Pakistan and has pharmaceutical importance [28]. When basil seeds are soaked in water and the mucilage content is high, they gain a gelatinous appearance. This high mucilage content of basil seeds can make it a novel source of edible gum [29]. Basil seed mucilage is squeezed out either by alcohol precipitation or via cold-water extraction. This mucilage contains uronic acid (±6.5%) and has a hetero-polysaccharide structure. This mucilage is responsible for forming an acidic anionic gel, when coming into contact with an aqueous solution. Two major components ((i) acid-stable core glucomannan (43%) and (i) glucose to mannose 10: 2 plus (1 – →4) linked xylan (24.29%)) are found in basil seed gum [30,31,32,33]. Recently, a few studies have been carried out on smearing BSG, a novel polysaccharide having sustained properties due to its super-absorbent nature. It is also used as a coating material in cephalexin-linked nanoparticle formulation by a magnetic field using a Fe_3_O_4_-based targeted technique [34]. Moreover, BSG is a natural hydrogel, having multiple properties, such as water swelling and thickening. These properties made it a desired option for the production of hybrid hydrogels [35]. Usually, hydrogels of a polysaccharide nature lack sensitivity properties and not respond to pH and temperature-type environmental stimuli, limiting their use as NDDS [36]. These problems can be overcome, using different monomers by copolymerization of polysaccharide.

Captopril is used for the treatment of heart diseases (hypertension and congestive heart failure), as a competitive inhibitor of ACE (angiotensin I-converting enzyme), by converting the angiotensin I to angiotensin II [37]. This drug has molecular weight (217.29 mol/g) and after an oral dose of captopril, its bioavailability is around 60–75%, due to a short half-life (1–2 h) and peak blood-level absorption around 1 µg/liter within 30 to 60 min by using a dose of 100 mg. Its short half-life is one of the most important drawbacks of captopril, requiring successive administration of the drug, which prompts poor patient compliance [38].

By using two different monomers, such as acrylic acid (AA) and *N*, *N*-Methylenebisacrylamide (MBA), along with the pH-sensitive polymeric matrix of basil seed gum (BSG), hydrogels were prepared using a free radical copolymerization technique using potassium per sulfate (KPS) as a cross linker. This polymeric technique was used with an aim to extend the release of the captopril drug. The reaction mechanism of copolymerization is shown in Figure 1, where potassium per sulphate (KPS), an initiator, generated initial free radicals upon heating (70 °C) that reacted the hydroxyl of basil gum and vinyl group of acrylic acid to form macro initiators. These reactive molecules then copolymerized in the presence of MBA to form a cross-linked BSG-AA copolymer network. After formulation of hydrogels by using different ratios of polymer and monomers, we examined the effect of polymeric composition on the release patterns of captopril in acidic and phosphate-buffer solutions, respectively. Swelling properties were evaluated at both pHs. Drug-release patterns and mechanisms were also analyzed at in vitro level from the polymer matrix. Next, formulated cross-linked hydrogels were evaluated to see the distinctive behavior of chemical compounds and new bond formation through FTIR spectroscopy. Other techniques, such as differential scanning calorimetry (DSC), X-ray diffraction (XRD), scanning electron microscopy (SEM) and acute toxicity studies, were carried out on the formed hydrogel. Chemical structures of the polymeric matrix of basil seed gum (BSG), with two different monomers, acrylic acid (AA) and *N*, *N*-Methylenebisacrylamide (MBA), are shown in Figure 1.

## 2. Experimental

### 2.1. Materials

Captopril (Warrick Pharmaceuticals Pvt. Ltd., Islamabad, Pakistan) and basil seeds were used for the extraction of basil seed gum; the molecular weight and intrinsic viscosity of BSG were 2320 kDa and 11.38 dL/g, respectively. Potassium per sulphate (KPS) as initiator and acrylic acid (AA) as monomer were procured from Sigma-Aldrich, St. Louis, MO, USA. *N,N*-Methylenebisacrylamide (MBA) as cross-linker was obtained from Fluka (Buchs, Switzerland). All chemicals used were of analytical grade.

### 2.2. Basil Seed Gum Extraction and Purification

First, we rinsed the seeds with a given volume of water in a short time and mixed with water (at the given water/seed ratio) at 60 °C at a speed of 150 rpm for 30 min. The gum layer on the surface of the seed was scraped off by passing the seed through an extractor with a rotating rough plate, thereby separating the gum from the expanded seed. Then, the separated gums were collected, sticking the remaining gums to the seeds and immersing them in water in a rotary extractor. This process was carried out four times. The collected gum was then filtered and dried using a vacuum oven at 50 °C. Finally, the dried gum was ground up, put in a plastic bag and stored in a cool and dry place. In order to purify the extracted gum, we added 3 volumes of 95% ethanol to 1 volume of extracted gum and left it for 30 min. A sieve was used to recover the precipitate to drain the excess solvent. The final precipitate was then dried in a vacuum dryer at 40 °C [39]. After drying, BSG was gathered, ground with dry blender, sieved through # 80, and kept in a sealed plastic container for further utilization.

### 2.3. Basil-Seed-Gum-Based Hydrogels Fabrication

BSG-co-poly (AA)hydrogels were prepared through various feed composition using free radical copolymerization technique [40], using BSG (as polymer), AA (as monomer) and MBA (as crosslinker) in different feed ratios. Composition of all prepared formulations is described in Table 1. Briefly, given amount of BSG added in distilled water was kept stirring overnight until completely dissolved. Acrylic acid was added drop wise to the solution, and then the KPS aqueous solution was added to the mixture with continuous stirring for 15 min to ensure complete mixing. Finally, MBA (dissolved in distilled water) was added slowly to previously made solution until homogeneous solution was formed. We poured the final solution into glass test tube and transferred to a preheated water bath at 55 °C. Temperature was gradually raised up to 70 °C. After 24 h, the tube was removed from water bath and cooled to room temperature. We removed the gel from the tube and cut into 1 cm wide discs with a sharp blade. We then washed the discs with ethanol–water (40:60) mixture for the removal of impurities, until the pH of washing mixture became constant. Finally the discs were kept for drying in hot-air oven at 40 °C [41].

### 2.4. Characterization

#### 2.4.1. Fourier Transformed Infrared (FTIR) Spectroscopic Studies

Dried discs of formulated hydrogels were crushed into fine powder. Sample with ratio of 1:100 was mixed with potassium bromide matrix and pressed to form pellet under pressure of 65 kN for 120 s using pressure guage. For low absorption of IR, pellet should not be opaque. We allowed them to dry at 40 °C to remove moisture content. FTIR spectrophotometer (IR prestige-21 Shimadzu) recorded spectra; wavelength range of 4500–500 cm^−1^ was used [40,42].

#### 2.4.2. Differential Scanning Calorimetric (DSC) and Thermo Gravimetric Analysis (TGA)

Thermal analysis of captopril, BSG, acrylic acid (AA), unloaded and loaded hydrogels was conducted using thermo gravimetric analyzer (TGA, NETZSCH Company, Waldkraiburg, Bavarian State, Germany). For conducting thermal analysis samples were heated using rate of 10 °C/min as of 0 to 500 °C under nitrogen stream [43,44].

#### 2.4.3. Powder X-ray Diffraction (PXRD)

PXRD was performed to check the properties of the synthesized hydrogel. The PXRD style is (X’Pert Pro via PANAnalytical), using the latest Ni-filtered CuKα radiation (1.314 A°) with a voltage of less than 40 kV, with a scan charge performed 1° in step with min over 10° to 40° diffraction perspective (2q) range. The XRD patterns of pure drug, loaded and unloaded hydrogels were carried out, and before scanning, the dried hydrogel was converted to a fine powder.

#### 2.4.4. Scanning Electron Microscopy (SEM)

Scanning electron microscope was used to examine the surface morphology of hydrogels (Hitachi, S 3000 H, Japan). SEM samples were arranged on an aluminum mandrel, and then gold was sputtered using palladium. Imaging was conducted at an operating range of 10–25 mm with a 10 kV accelerating voltage.

#### 2.4.5. Porosity Studies

Porosity studies were performed by solvent replacement method. Weighed dried discs (R1) of all formulations of BSG-co-poly (AA) hydrogels were soaked in absolute ethanol for 48 h. Hydrogel discs were taken after 48 h; excess solvent was removed by blotting with tissue paper and we weighed the discs accurately (R2). Diameter and thickness of disc were also determined [45]. Porosity % was calculated by using the following equation.
$$\mathrm{Porosity}\;\%=\frac{R2-R1}{\rho V}\;\times \;100$$

ρ is density of absolute alcohol, V is the volume of hydrogel after swelling.

#### 2.4.6. Rheological Properties of BSG-co-poly (AA) Hydrogels

The rheological properties were evaluated by using a rheometer (Gemini 200-Malvern Instruments, Malvern, UK) fitted with plate and cone geometry. Cone–plate geometry having angle of 4 ° and diameter of 20 mm to 40 mm was used in the following experiments. Four types of different shear tests were applied [46].

Time sweep tests: Following tests were carried out to obtain difference between the elastic and viscous nature of the gel involved in gelation time for more than 1 h. The samples were subjected to constant strain below the critical range (i.e., 1%) and at constant stress of 1 Hz and finally G’ and G” were evaluated. G’ represents the elastic part, G” represents the viscous part. Both of these elastic and viscous responses are 90° apart from each other. Materials having both the elastic and viscous response that is between 0° and 90° are known as viscoelastic [46].

Frequency Sweep Tests: After completion of time sweep test, i.e., around 80 min, Frequency sweep tests were carried out at a constant strain between 0.01 and 10 Hz, at fixed hydrogel temperature, i.e., 37 °C. Again, the elastic (G’) and viscous (G”) modulus were determined [47].

Temperature Sweep Test (heating–cooling cycle): This test was performed to see the thermal behavior of BSG-co-poly (AA) hydrogels at constant frequency of 1 Hz and strain within linear viscoelastic range, and the temperature was increased from 5 °C to 90 °C with a heating rate of 2 °C/min. Stability of hydrogel was checked by keeping the temperature higher than 80 °C for 30 min at physiological pH (pH = 7).

Flow Behavior (viscosity vs. shear rate): The purpose of this test was to analyze the effect of varying shear rates on the gel viscosity with different concentrations. The flow curves (viscosity vs. shear rate) for different BSG-co-poly (AA) hydrogel formulations were drawn to evaluate the impact of varying concentrations of monomers, such as acrylic acid (AA) [46].

### 2.5. Determination of Gel%, Yield% and Gel Time

Calculating the gel% and yield% in the synthesis of BSG-co-poly (AA) hydrogels can reveal the polymerization content of the reactants. The hydrogels were dried to a consistent weight (mi) in a vacuum oven before being soaked in water for 7 days with intermittent shaking to eliminate the water-soluble moieties. The water-insoluble part of the hydrogel was then dried to a consistent weight in a vacuum oven (md). The gel% and yield% were calculated as follows.
$$\mathrm{Gel}\%=\frac{\mathrm{md}\;}{\mathrm{mi}}\;\times \;100\;\mathrm{Yield}\%=\frac{\mathrm{md}}{\;\mathrm{mc}}\;\times \;100$$
where mc is combined weight of the developed formulation reactants.

### 2.6. Swelling Studies

Dynamic and swelling studies were performed to assess the swelling behavior of BSG-co-poly (AA) hydrogels.

#### 2.6.1. Dynamic Swelling Studies

The dynamic swelling study was carried out at 37 °C in 100 mL of 0.1 N hydrochloric acid (pH 1.2) and phosphate buffer (pH 7.4) [40]. The pre-weighed discs were immersed in their respective media and weighed after blotting them dry with a filter/tissue paper at regular intervals, up to 72 h. We used the following formula to calculate the dynamic swelling rate [48].
*Q* = W_s_/W_o_

#### 2.6.2. Equilibrium Swelling Studies

For the determination of equilibrium swelling studies, the hydrated disc was kept in the respective medium for 15–20 days [40]. The equilibrium–swelling ratio was calculated using the following formula.
S _(Eq)_ = W_h_/W_d_
W_h_ = weight of swollen gel at time t
W_o_ = initial weight of dry gel

### 2.7. Drug Loading

The swelling–diffusion method was used to assess the captopril content in BSG-co-poly (AA) hydrogels. The BSG-co-poly (AA) hydrogel discs were submerged in 100 mL captopril (1%) in phosphate buffer pH 7.4 for 30 min. The discs were withdrawn after 24 h and left to dry at ambient temperature until being dried in a 40 °C oven. The drug loading was determined by extracting a calculated amount of the drug with the same solvent that was used to load it. Using 25 mL of 0.2 M fresh buffer, the drug was recovered from the BSG-co-poly (AA) hydrogel disc until no drug was detectable in the extraction solution. A calibration curve of drug dilutions in 0.2 mL was used to calculate the quantity of acquired drug on disc using a UV-visible spectrophotometer (UV-1601Shimadzu, Shimadzu, Kyoto, Japan) [49].
Amount of loaded drug = (W_d/_W_o_)
W_d_ = weight of loaded drug disc after drying
W_o_ = Initial weight of unloaded drug disc

### 2.8. In Vitro Drug-Release Studies

The BSG-co-poly (AA) hydrogel was examined using an automatic USP device-II (paddle method) with an autosampler (Watson Marlo, Stockholm, Sweden) to determine the drug-release pattern. A dissolution medium containing 0.1N HCL pH 1.2 and pH 7.4 phosphate buffer at 37 °C ± 5 °C and 100 rpm was used to study the release of captopril from the hydrogel tray for 24 h. We regularly sampled and checked the UV-Vis spectrophotometer (UV-1601 Shimadzu) at λ_max_ 205 nm [43].

### 2.9. Drug-Release Kinetics

We used various kinetic models, i.e., zero and first order, Higuchi and Korsmeyer–Peppas models, to imitate the various drug-release mechanisms from BSG-co-poly (AA) hydrogels.

### 2.10. Sol–Gel Analysis

Sol–gel analysis of formulated hydrogels was carried out to examine the soluble uncross-linked and insoluble cross-linked parts of the hydrogel. Sol fraction refers to the soluble uncross-linked portion, while gel fraction refers to the insoluble cross-linked portion. For sol–gel analysis, a Soxhelt extraction technique was used. We weighed the hydrogel discs and placed in a round-bottom flask holding a precise amount of deionized distilled water. A condenser was linked to the flask’s spherical bottom. The Soxhelt extraction was performed at 85 °C for 12 h. The isolated hydrogel disc was then placed in a vacuum oven at 40 °C until completely dehydrated. Hydrogel discs were then dried and weighed again. Sol–gel fraction was calculated by using following equation.
$$\mathrm{Sol}\;\mathrm{Fraction}\;\%=\frac{R\;1\;–\;R\;2}{R\;1}\;\times \;100$$
Gel Fraction = 100 − Sol Fraction

### 2.11. Statistical Analysis

Graph pad prism 5.00 (Graph Pad Software Inc., La Jolla, CA, USA) was used for statistical analysis. The R^2^ values for the accuracy analysis as well as the predictability of different kinetic models were evaluated using DD Solver software, 1.0 (Trial version).

### 2.12. Acute Toxicology Study

The acute oral toxicology study of BSG-co-poly (AA)-captopril-loaded hydrogel was evaluated by using the MTD (Maximum Tolerated Dose) method. Swiss albino mice (25–30 g) of both sexes were purchased from University of Sargodha (UOS), animal laboratory. All experiments were performed according to OECD guidelines, and further verified by the ethics committee of UOS (Ref. No. 108–2020/PREC). Two groups (control and treated) having an equal number of male and female mice (*n* = 8) were kept under 12 h light/dark cycle with feeding supply of standard diet plus water in clean housed facility. BSG-co-poly (AA)-captopril-loaded hydrogels (150 mg/kg) were given orally by an intubation cannula to treated group only [50]. All details linked to indication of toxic effect, bad health, mortality and any other activity effect were observed in all animals for two weeks, twice daily. Blood, clinical biochemistry, gross necropsy and histopathological analysis were carried out after two weeks. Overall, treated group was compared with controlled group for each parameter.

Plasma was separated and analyzed for different clinic-pathological investigations. ALT, AST, cholesterol, triglycerides, creatinine, urea and uric acid were examined. After 2 weeks, all mice were sacrificed. We performed gross autopsy on different important organs such as the heart, liver, spleen, kidney and stomach. We then weighed the organs and calculated the relative organ weights. All organs were stored in 4% buffered formalin solution. Paraffin embedding was done and selected tissues were cut into 4–5 µm thickness to perform histopathology analysis by hematoxylin–eosin staining.

## 3. Results and Discussion

### 3.1. Physicochemical Analysis of Prepared Hydrogels

#### 3.1.1. Fourier Transformed Infrared (FTIR) Spectroscopic Studies

FTIR spectra of BSG, BSG-co-poly (AA) hydrogel and captopril are shown in Figure 2. BSG-linked spectroscopic analysis showed an OH group at 3481.51 cm^−1^, C-H in CH_2_ at 2995.45 cm^−1^, COO (asymmetric and symmetric stretching) due to uronic acid at 1689.64 cm^−1^ and 1535.34 cm^−1^, C-O at 1228.66 cm^−1^, and C-O-C stretching at 970.19 cm^−1^ (a). FTIR of captopril (b) showed characteristic peaks at 962.48 cm^−1^ (CN group) cm^−1^, 1587.42 cm^−1^ (nitrogen ring), 1743.65 cm^−1^ (Carbonyl group) 639.49 cm^−1^ (COO-ion) and 2565.33 cm^−1^ (SH group), as reported previously. BSG-co-poly (AA) hydrogel (c) shows similar peaks of hydrogel, along with an additional peak in the carbonyl group at 1732.65 cm^−1^, which may represent the formation of an ester link between acrylic acid and gum molecules. An additional peak at 3572–3645 cm^−1^ appeared, which could be due to the N-H of amide bonds present in the cross linker MBA [51]. Captopril-loaded hydrogel showed characteristic peaks of captopril, indicating no interaction between the hydrogel components and drug.

#### 3.1.2. TGA (Thermo Gravimetric Analysis) and DSC (Differential Scanning Calorimetric)

DSC and TGA analysis were carried out to test the thermal stability and confirm cross linking. The results are shown in Figure 3. In the TGA curve of pure captopril, the peak of initiation of degradation was detected at 274.9 °C (Figure 3a A). TGA revealed the fact that basil seed gum (Figure 3a B) showed an endothermic peak between 276.32 and 377.37 °C with 13.88–52.89% weight loss [52]. Another broad peak appearance from 97 °C to 160 °C was seen in BSG-co-poly (AA)-loaded hydrogel (Figure 3a D). This information highlights that captopril did not change its thermal behavior and molecularly dispersed in different hydrogel matrices. Both drug-loaded and drug-unloaded (Figure 3a C,D) -linked major endothermic peaks were seen at 240 °C. The DSC of the pure drug (captopril) (Figure 3b A), with a sharp peak appearance at 106 °C, represents the melting point of the drug [53]. The thermal stability of the formulated hydrogel was improved, as evidenced with the TGA of BSG, as shown in Figure 3a. Greater thermal stability of the manufactured hydrogels indicated a strong interaction of components because of cross linking, grafting and copolymerization.

#### 3.1.3. Powder X-ray Diffraction Analysis

Captopril and developed hydrogels were also evaluated by XRD to find out the crystalline and amorphous structure. The sample was ground to uniform size and evaluated, and results are shown in Figure 4. Captopril is crystalline in nature, as shown in Figure 4; however, hydrogels prepared by cross linking using polymers generally produced amorphous structures with denser peaks. XRD patterns of captopril showed distinct, intense and sharp peaks at 2Ɵ and were obtained at 11°, 18°, 28°, 34°, 22°, with peak intensities of 200 and more, 300. At 20°, the intensity of the peak was maximum (1000 at 26° peak intensity 600), which highlights the crystalline nature of the drug (captopril). In BSG-co-poly (AA)-loaded hydrogels, captopril was entangled and distributed in formulation; a distinctive sharp peak at 2Ɵ = 27.790 of pure captopril was much reduced and broadened in BSG-co-poly (AA) hydrogel, as shown in Figure 4. The combined effect showed amorphous behavior as the crystalline property of captopril diminished due to grafting with basil mucilage and AA, thereby improving solubility [54].

#### 3.1.4. Scanning Electron Microscopy

Microscopic analysis was taken at 100 µm, and the cross section at 50 µm, as shown in Figure 5a,b. This analysis showed a compressed gel with an irregular, uneven and rough surface, which helps to mediate water uptake promptly. These characteristics, in turn, increase the swelling capacity of the gel and the drug release. The porous architecture of hydrogels not only increases the surface area but also the water uptake. This behavior is directly linked to the polymeric-network-based swelling tendency [55]. Therefore, SEM analysis showed the surface roughness is mainly responsible for the entrapment of drug/solute particles in the hydrogels.

#### 3.1.5. Porosity Studies

Porosity refers to pore volume, pore size and number of pores present in the polymer network. Hydrogel porosity has a substantial impact on the drug delivery process. The pore size is significantly affected by the hydrogel composition [45,56]. The results of porosity of BSG-co-AA hydrogels are displayed in Figure 6. According to the graph, hydrogel porosity decreased with increasing BSG and MBA content. Increased cross-linked density leads to a reduction in pore size [57,58]. Higher AA content generated hydrogels with higher porosity, which is associated with higher swelling of the hydrogels, contributed by an increasing amount of hydrophilic monomer. Similar results were reported for AA-co-AM super-porous hydrogels [59].

#### 3.1.6. Rheological Properties of BSG-co-poly (AA) Hydrogels

These tests are required to understand the fundamental properties of hydrogels, such as physical strength, cohesiveness, elasticity and rubbery behavior, etc. During the time sweep test, time-dependent changes in the G’ and G” of the hydrogels were examined at the constant frequency (1 Hz) and room temperature (37 °C). The value of G’ was higher than the value of G”, as shown in Figure 7a, showing more elasticity in BSG-co-poly (AA) hydrogels [8].

After the time sweep experiment that was conducted at 37 °C after 80 min, we performed a frequency sweep test. By increasing the frequency, no more changes in values of G’ and G” of the hydrogels were seen, as in Figure 7b. The value of G” of the hydrogels was much lower than G’, indicating the stability of the BSG-co-poly (AA) hydrogels in their swollen states [60].

During the temperature sweep test, the BSG-co-poly (AA) hydrogels solution at the pH = 7 i.e., physiological pH, converted to a gel state. The sharp rise in G’ (Elastic modulus) was seen in the heating curve (b) with the increase in temperature, which highlights the gelation process BSG-co-poly (AA) hydrogels solution, while the decrease in the G′ in the cooling curve (a) highlights a certain thermos-reversibility process (a tendency of the gel state conversion to the liquid state), as shown in Figure 7c [47].

A flow behavior test (viscosity vs. shear stress rate) was carried out at room temperature for three BSG-co-poly (AA) hydrogel formulations, with varying concentrations of acrylic acid (AA). The shear stress profiles of all three formulations of BSG-co-poly (AA) hydrogels revealed the thinning of gels. The slope of the shear stress vs. viscosity was observed to decrease by increasing the shear rate, as in Figure 7d. Apparently, the viscosity of all three formulations of BSG-co-poly (AA) hydrogels were comparable with an increased concentration of monomer. This result was in accordance with the gelation time study; as we already mentioned, viscosity was increased by decreasing gelation time due to an increase in concentration of monomers [46].

### 3.2. Determination of Yield and Gel Percentage and Gelling Time

The efficiency of gelation was also an important parameter. The influence of wt.% of BSG, AA and MBA on yield and gel percentage and time for gel formation is shown in Figure 8. Gel% and yield% were directly linked to the amount of reactants. With an increasing amount of polymer, more active sites for cross linking were available and increased gelation occurred. Hence, gel time also increased. Figure 8A displays the increase in yield and gel percentage and time for gel formation, when BSG parts in the formulation were increased from 0.5 to 1.0 wt.% [61].

The effect of monomer AA concentration is described in Figure 8B. The yield and gel% were increased via increasing the monomer (15–35 wt.%). A high concentration of reactive double bonds led to a dense network and gel formation. Conversely, the gel time decreased with the increase in the amount of monomer because of the high reaction rate [62].

Figure 8C displays the effect of the increase in the cross-linker amount from 0.2 to 0.6 wt.%. Due to the presence of a high amount of cross linker, we observed the formation of a more densely cross-linked network. Therefore, higher values for gel% and yield% were obtained. However, gel time decreased because more reactant was present for the same amount of monomer and polymer [63].

### 3.3. Swelling Behavior of Hydrogels

The water intake of initially dry gels was monitored for 72 h and measurements were taken until each hydrogel sample attained a consistent weight. For a particular hydrogel sample, this constant value was used as the equilibrium swelling. The swelling behavior of BSG-co-poly (AA)-copolymerized hydrogels was explored, depending on pH, BSG content, AA content and MBA content.

#### 3.3.1. Effect of Medium pH on Hydrogel Swelling

The pH responsiveness of hydrogels containing ionizable groups was entirely determined by composition. The effect linked to pH on swelling of BSG-co-poly (AA) hydrogels was investigated at pH 7.4 (phosphate buffer) and HCl (0.1 M) with pH 1.2. Figure 9A shows the dynamic swelling behavior of BSG-co-poly (AA) hydrogel (formulation CH5) over a time of 72 h, at low (1.2) and high (7.4) pH. It is obvious from the graph that at pH 7.4, the hydrogel swelled significantly more than at pH 1.2. Acrylic acid is an ionizable molecule. The ionization of the carboxyl group was affected mainly by the pH of the surrounding medium and pH has a significant impact on the swelling behavior of hydrogels. Acrylic acid has a pKa around 4.5 [64]. When the medium pH level was below the pKa value, the carboxylic groups were totally unionized. Hydrogen bonding among carboxylic molecules was maximum, whereas the swelling degree was relatively minimal. At pH values higher than the pKa value, the carboxylic groups become ionized (COOH was converted to COO^−^) and ended up breaking the hydrogen bonding. The electrostatic repulsion of the ionic charges in the hydrogel’s network caused it to swell. At pH 7.4, the COO^−^ − COO^−^ anionic repulsions became significant, leading to a marked increase in polymer network expansion and swelling [65], and a similar effect of external pH was shown as semi-interpenetrating polymer network (IPN) hydrogels made by Guar gum-poly (acrylic acid) [66].

#### 3.3.2. Effect of BSG Content on Hydrogel Swelling

The swelling forces of a polymer containing ionic groups were substantially increased by charging sites on the polymer chains [67]. BSG possessed many of the free ionic groups (hydroxyl and carboxylic) in its structure that divulge a hydrophilic character to the polysaccharide. BSG has the ability to absorb a significant amount of water, which suggested the high swelling and water-holding capacity of mucilage [68]. How BSG content influenced the swelling behavior of BSG-co-poly (AA) hydrogels is shown in Figure 9B. To investigate the BSG content effect, the amount of BSG in the formulation composition was varied from 0.5 to 1.0 wt.%, keeping the amount of other ingredients constant. The concentration of BSG was selected based on the solubilty of BSG polymer and viscosity. With an increasing amount of BSG, the hydrogel swelling in PBS (pH 7.4) was found to increase as expected. The diffusion of water into the hydrogel network due to the presence of hydrophilic molecules, as well as electrostatic repulsion between ionized carboxylic groups (COO^−^) at neutral pH, resulted in an expansion of the hydrogel network. Hence, equilibrium swelling values were increased from 67.2 to 75.9 g/g; an increase in swelling with increasing polymer content for sodium alginate/acrylamide cross-linked hydrogels has been also reported [69].

#### 3.3.3. Effect of AA on Swelling of Prepared Hydrogel

The changes in swelling of the prepared hydrogels with varying AA content were investigated using 15–35 wt.% AA. The results are displayed in Figure 9C. The resulting graph is evident about the swelling ratio seen at pH 7.4, showing that it was greatly influenced by a high concentration of AA. Increasing the AA content in the hydrogel network has been associated with an increase in the grafting and water-uptake capacity of hydrogels. A high monomer concentration provides a higher number of reacting molecules at the reaction site that form the polymeric network. The cavities created inside hydrogels due to network formation hold water molecules upon absorption. Further, a higher AA concentration in the composition leads to an increased number of ionized carboxylic groups (COO^−^) in AA molecules [70]. The electrostatic repulsion among neighboring COO^−^ forces expansion of the polymeric network resulted in high swelling and water uptake. When the AA content in the BSG-co-poly (AA) hydrogels was increased from 15 to 35 wt.%, the equilibrium swelling ratio increased from 36.4 to 79.7 g/g. Therefore, AA content can serve to control the swelling behavior of BSG-co-poly (AA) hydrogel.

#### 3.3.4. Effect of MBA Content on Hydrogel Swelling

To observe the effect of MBA (cross linker) content on hydrogel swelling, the amount of MBA was varied from 0.2 to 0.6 wt.%. Figure 9D illustrated the change in equilibrium swelling due to varying amounts of MBA. Lower swelling ratios were obtained with an increased MBA content. When the amount of cross linker decreased, while the amounts of monomer and polymer were constant, more crosslinks were formed. Cavities with smaller mesh size were formed with reduced water-holding capacity. The enhanced crosslink density restrains the expansion of the hydrogel network. As a result, the swelling ratio was decreased, and it is also explained that an increase in the cross linking of hydrogels hindered the mobility of the polymeric chain, hence, lowering the swelling ratio [71].

### 3.4. Drug Loading

Drug loading was a critical aspect in evaluating the potential of hydrogels as possible drug carriers. An ideal drug carrier should be able to load a sufficient quantity of drug molecules inside its network to reduce the number of doses administered [72]. The amount of captopril in milligrams loaded in all formulations of BSG-linked co-poly (AA) hydrogels, prepared with multiple wt.% of BSG, AA and MBA, is shown in Figure 10A–C. From the results, it can be inferred that the overall drug-loading trend was consistent with the findings of the swelling behavior of these hydrogels. Formulation CH6 containing maximum AA wt.% showed the highest swelling and captopril loading. In general, the amount of loaded captopril increased with increasing BSG and AA content. On the other hand, the drug loading of hydrogels decreased when MBA content was increased. Formulation CH9 containing maximum MBA wt.% showed minimum swelling and captopril loading, describing that drug loading capacity in hydrogels is directly proportional to swelling capacity [73]. The greater the swelling, the greater the amount of the drug loaded in the hydrogel and vice versa; this has also been reported before about increased drug-loading behavior by increasing the polymer amount, and a decrease in drug loading with an increased amount of MBA [74].

### 3.5. In Vitro Drug-Release Studies

#### 3.5.1. Effect of pH on Captopril Release from Hydrogels

The effect of the used medium (different pH) on drug release was investigated. The release studies of captopril-loaded BSG-co-poly (AA) hydrogel (formulation CH6) were conducted in 0.1 M HCL solution (pH 1.2) and PB (pH 7.4). As evident from Figure 11A, the drug release followed swelling behavior.

#### 3.5.2. Effect of Different Composition Parameters on Captopril Release from Hydrogels

Release patterns of captopril in PB (pH 7.4) from BSG-co-poly (AA) hydrogels containing varying contents of BSG, AA and MBA are shown in Figure 11B–D. As shown in the Figures, initially, the rate of drug release was high, but it reduced gradually over time. This behavior can be explained as follows. Firstly, the initial release in burst form was due to drug molecules being trapped on the surface of the hydrogel. Secondly, in the beginning, the drug concentration gradient was high due to the low concentration of drug in the release medium. Through the releasing phenomenon, the drug concentration in the medium increased. Consequently, the concentration gradient decreased and the drug was released at a slower rate. Hence, we can conclude that the driving force exerted due to this concentration gradient is responsible for captopril release from BSG-co-poly (AA) hydrogels. Initially high and then slow diffusion of drug molecules from hydrogel matrices in the drug release study of chitosan-acrylamide cross-linked hydrogels has been demonstrated before [75]. It was also observed that drug release followed the swelling properties of hydrogels. Like swelling, increasing the amount of BSG and AA and decreasing the amounts of MBA resulted in an increased % captopril release. Similar hydrogel-based release profiles have also been reported in different studies [76,77].

### 3.6. Drug-Release Kinetic Models

To find out the drug release kinetics or patterns, dissolution data were fitted and seen in kinetic models, as shown in Table 2. Drug release behavior mostly relates to the Korsmeyer–Peppas model that exhibits a correlation coefficient (R^2^) > 0.98 for hydrogel formulations. The diffusion coefficient “*n*” in the Korsmeyer–Peppas model demonstrates that if 0.45 ≤ *n*, then the release pattern is according to Fickian diffusion, and if 0.45 ≤ *n* ≤ 0.89 was found, then the release pattern is according to non-Fickian diffusion. Further, if *n* = 0.89, then the release pattern is a Case II transport (typical zero-order) and *n* ≥ 0.89 indicated the release pattern as super Case II transport. If “*n*” value is within the 0.58–0.65 range, then the release pattern will be an anomalous/non-Fickian diffusion, representing the polymer relaxation and diffusion in a hydrated matrix.

### 3.7. Sol–Gel Analysis

The uncross-linked soluble component of hydrogels is called sol, whereas the insoluble cross-linked part is called gel. The gel fraction increased significantly as the amount of formulation materials, i.e., polymer, monomer, and crosslinker, increased. The results of the sol–gel analysis for different formulations of BSG-co-poly (AA) hydrogels is shown in Figure 12. With higher concentrations of BSG, AA, and MBA, the gel fraction improved. The polymerization process between BSG and AA raised as the number of available reactive molecules increased and, therefore, the gel fraction increased, and vice versa. Similarly, as the concentration of AA increased, so did the gel fraction. On the other hand, the proportion of gel to sol fraction was inverse. As the gel fraction increased, the sol fraction decreased. Thus, when the concentration of BSG, AA, and MBA increased, the sol fraction decreased [45,57].

### 3.8. General Signs of Acute Toxicology Study

After conducting this acute level study, no sign linked to mortality was seen in the trial period of 2 weeks, in both the control and treated groups. None of the appearance changes or signs of illness were seen in any mice during this observation phase. Body weight alterations were insignificant in both groups (Table 3). All vital organs, such as heart, liver, spleen, kidney and stomach, remained intact in both groups. After oral intake of BSG-co-poly (AA)-loaded hydrogels, its effect on food, water intake and body weight were also recorded, as shown in Table 3. Food and water intake was a little lower in the hydrogel-treated group, as compared to the control group (this may be due to hydrogel intake causing fullness of the stomach). Overall, food and water intake were casual in both groups, showing normal physiological behavior. In accordance with the globally harmonized system, LD50, higher than a 2000–4000 mg/kg dose for testing chemicals in Swiss albino mice and toxicity score was zero [78]. Therefore, our BSG-co-poly (AA) hydrogel formulation was in accordance with the abovementioned category, showing no signs of toxicity.

#### 3.8.1. Biochemical Plasma Analysis

Plasma is the best way to study the toxicity produced by chemicals under physio-pathological conditions. Plasma analysis of different chemical parameters linked to liver, lipid profiles and kidney functioning for both (control as well as treated group) are given in Table 4. Levels of transaminases (ALT and AST) in both groups appeared in the normal range and were comparable, i.e., 28–184 IU/L. Other than ALT and AST, lipid profiles (cholesterol and triglycerides level), creatinine, urea and uric acid levels also appeared in the normal range. These parameters represented no sign of disease or toxicity, neither in blood nor in liver nor kidney of both groups.

#### 3.8.2. Histological Study of Vital Organs

The microscopic tissue-level evaluation of both the control as well as the hydrogel-treated group was conducted. No pathological lesions were noticed in the heart, liver, spleen, kidney or stomach. The absolute organ weight for both groups was also in the normal range (Table 5). The heart tissue showed optimal integrity in the control group but a little vascular dilation was seen in the heart tissue in the treated group, which may be the effect of captopril on the heart vessels. Overall, no degenerative changes were seen in any of the mentioned vital organs, indicating they were free from any sort of significant pathology, as shown in Figure 13.

## 4. Conclusions

Highly porous BSG-co-poly (AA) hydrogels were formulated efficaciously via a technique dependent on free radical copolymerization for the sustained effect of captopril. DSC and TGA presented cross linking and made a stable hydrogel linkage, which was thermally more stable. Like swelling, increasing the amount of BSG and AA and decreasing the amounts of MBA resulted in increased % captopril release. Rheological studies indicated the stability of BSG-co-poly (AA) hydrogels in their swollen states. Elasticity dominated in our BSG-co-poly (AA) hydrogel systems (G’ > G”), both of low and high frequency. This corresponded to cross-linked hydrogel dispersions that exhibited no prominent change in G′ and G′′, being almost independent of oscillation frequency. The kinetics of drug release followed the Korsmeyer–Peppas model, which showed diffusion, linked to polymer relaxation and hydrated matrix. Acute toxicology studies of BSG-co-poly (AA) hydrogels revealed zero toxicity score. Finally, the results of the study led us to recommend that BSG- co-poly (AA)-based hydrogels offer a latent alternative to old dosage forms via the sustaining and controlling effect of drug delivery for extended periods.

## Figures and Tables

**Figure 1 gels-08-00291-f001:**
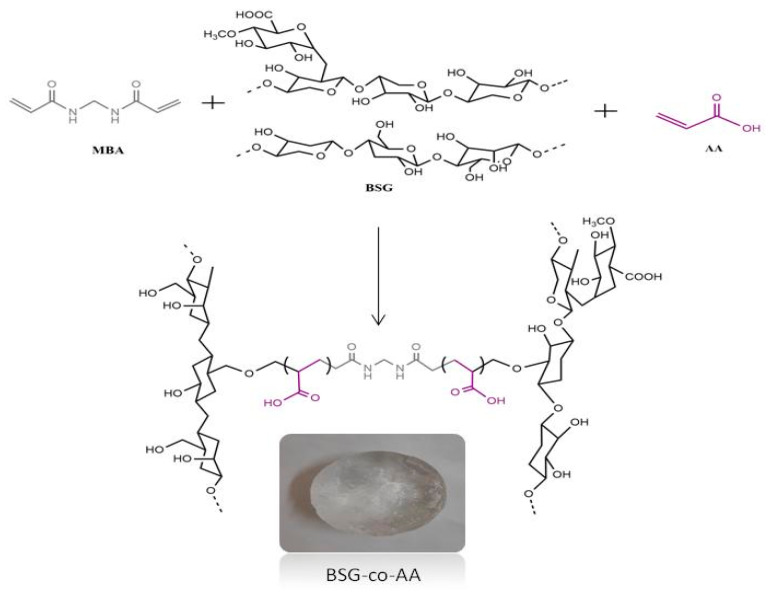
Chemical structure of polymer (BSG), monomer (AA), cross linker (MBA) and possible cross – linked structure of BSG–co–poly (AA) hydrogel.

**Figure 2 gels-08-00291-f002:**
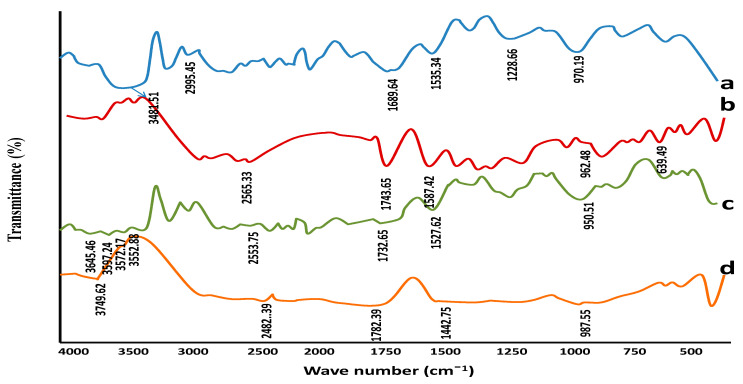
FTIR spectra of (a) BSG (b) captopril (c) BSG–co–poly (AA) unloaded hydrogels. (d) BSG–co–poly (AA) loaded hydrogels.

**Figure 3 gels-08-00291-f003:**
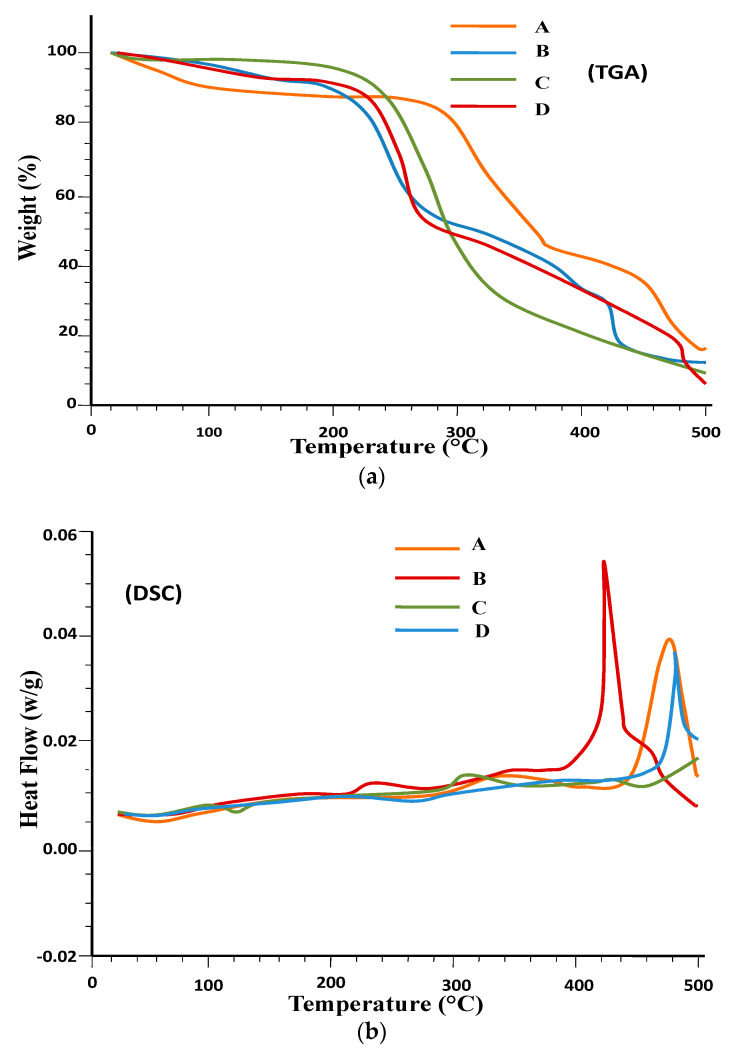
(**a**). TGA of (A) captopril; (B) BSG; (C) BSG-co-poly (AA)-unloaded hydrogel; (D) BSG-co-poly (AA)-loaded hydrogels, (**b**). DSC of (A) captopril; (B) BSG; (C) BSG-co-poly (AA)-unloaded hydrogel; (D) BSG-co-poly (AA)-loaded hydrogels.

**Figure 4 gels-08-00291-f004:**
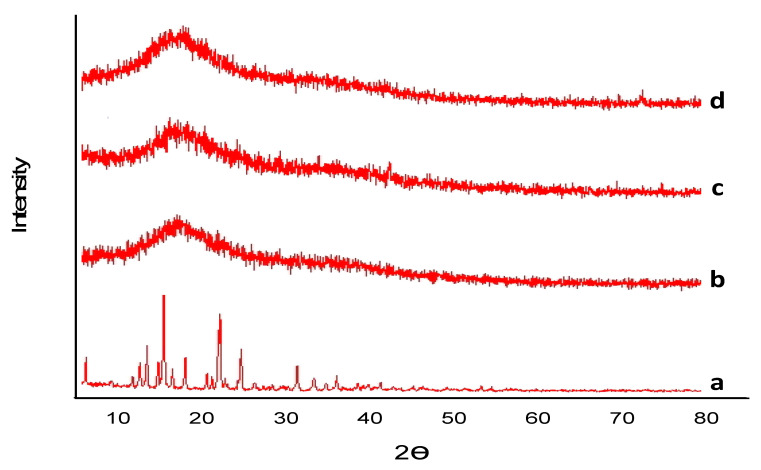
XRD of (a) captopril; (b) BSG-co-poly (AA)-unloaded hydrogel; (c) XRD of captopril (d); BSG-co-poly (AA)-loaded hydrogels.

**Figure 5 gels-08-00291-f005:**
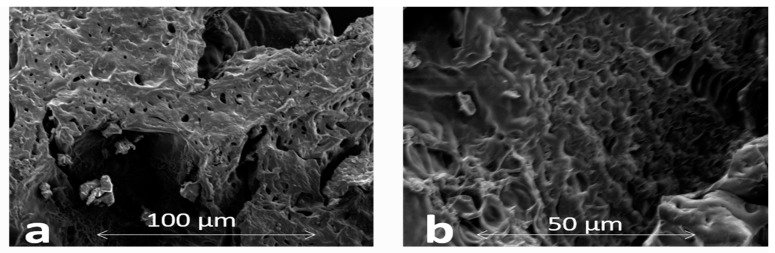
Surface (**a**) and cross-sectional (**b**) SEM images of BSG-co-poly (AA)-loaded hydrogel.

**Figure 6 gels-08-00291-f006:**
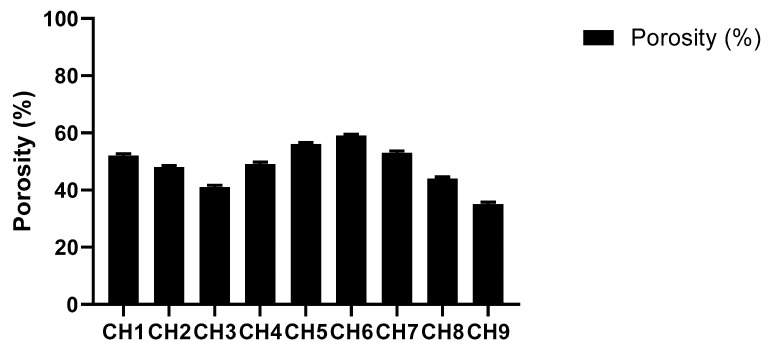
Porosity (%) of BSG-co-poly (AA)-loaded hydrogel.

**Figure 7 gels-08-00291-f007:**
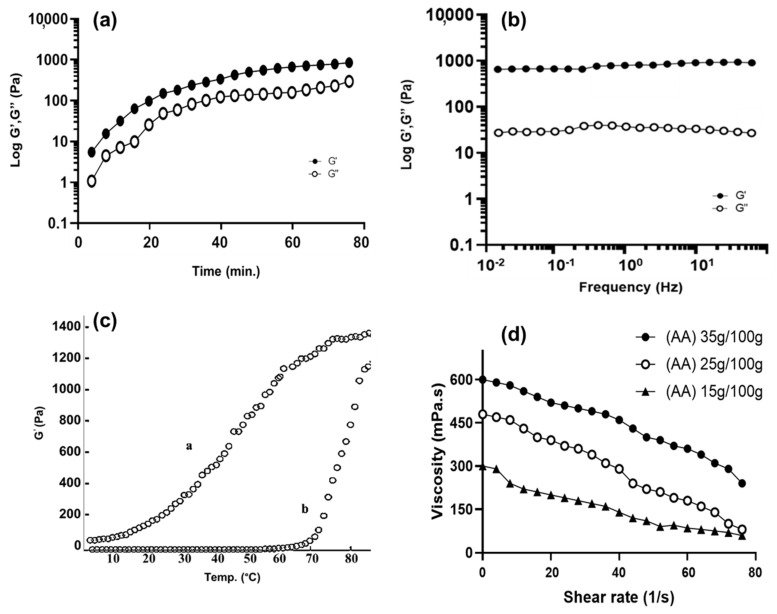
(**a**) Time sweep of BSG–co–poly (AA) hydrogels (**b**) frequency sweep of BSG–co–poly (AA) hydrogels (**c**) temperature sweep test of BSG–co–poly (AA) hydrogels solution (**d**) flow behavior test (viscosity vs. shear rate) of BSG–co–poly (AA) hydrogels.

**Figure 8 gels-08-00291-f008:**
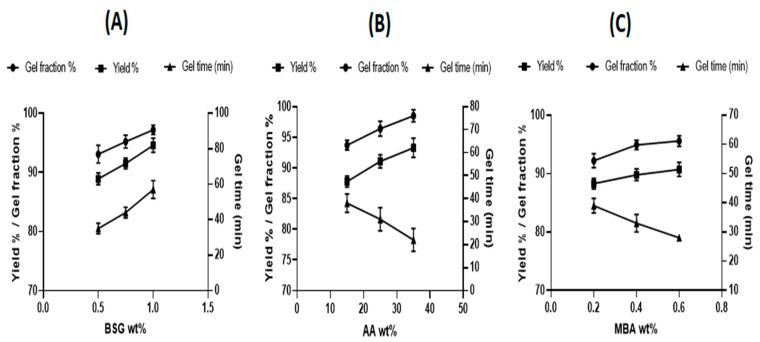
Effect of (**A**) BSG; (**B**) AA; (**C**) MBA on gel%, yield% and gel time.

**Figure 9 gels-08-00291-f009:**
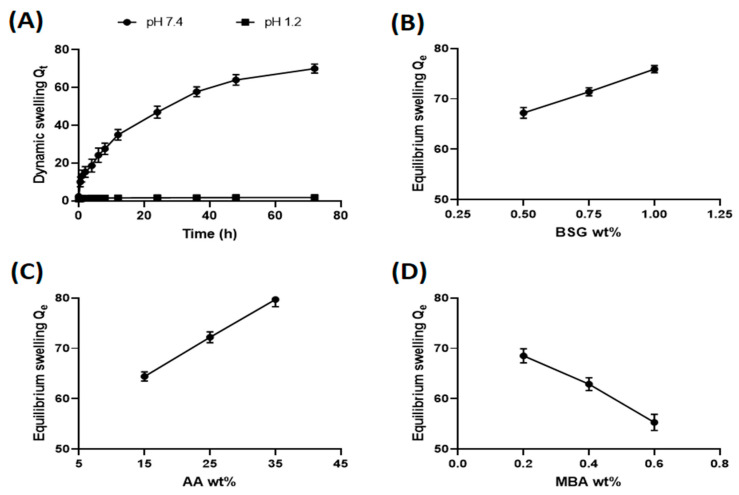
(**A**) Effect of different pH mediums on dynamic swelling; (**B**) effect of BSG, (**C**) effect of AA, (**D**) MBA wt.% on swelling.

**Figure 10 gels-08-00291-f010:**
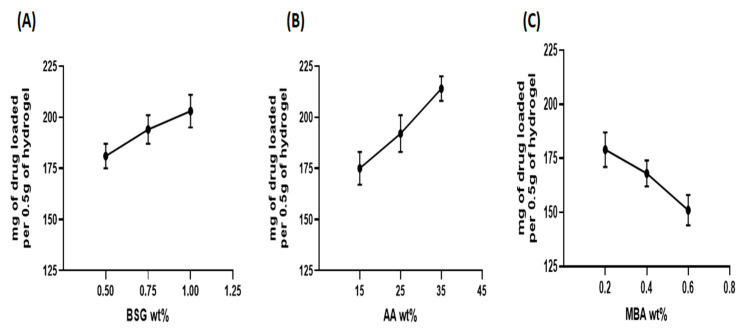
Effect of (**A**) BSG; (**B**) AA; (**C**) MBA on drug loading of hydrogel.

**Figure 11 gels-08-00291-f011:**
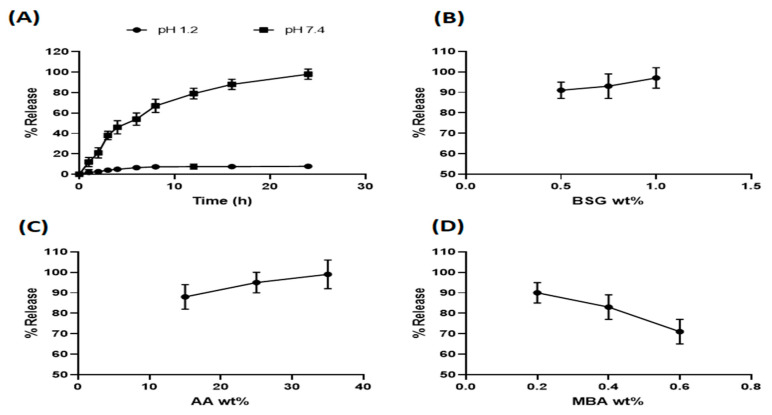
(**A**) Captopril release percent from BSG-co-poly (AA) hydrogels in acidic and basic medium; (**B**). Effect of BSG; (**C**) effect of AA; (**D**) MBA content on captopril % release.

**Figure 12 gels-08-00291-f012:**
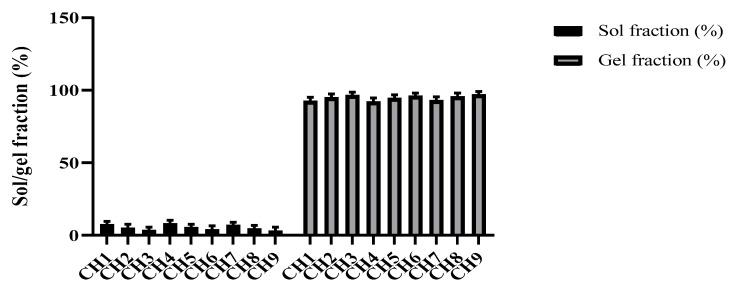
Sol–gel fraction % of BSG-co-poly (AA) hydrogels.

**Figure 13 gels-08-00291-f013:**
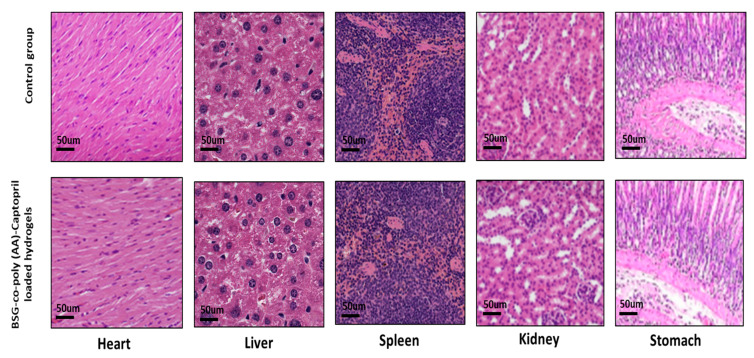
Histological analysis of control and BSG-co-poly (AA)-Captopril-loaded-hydrogel-treated group.

**Table 1 gels-08-00291-t001:** BSG-co-poly (AA)-captopril-loaded hydrogels formulations prepared by using different amount of polymer, monomer, co-monomer along with initiator.

Formulations	Polymer (BSG) %	Monomer (AA) %	Comonomer (MBA) %	Initiator KPS %
CH1	0.5	20	0.25	0.15
CH2	0.75	20	0.25	0.15
CH3	1	20	0.25	0.15
CH4	0.5	15	0.25	0.15
CH5	0.5	25	0.25	0.15
CH6	0.5	35	0.25	0.15
CH7	0.5	20	0.2	0.15
CH8	0.5	20	0.4	0.15
CH9	0.5	20	0.6	0.15

BSG: Basil seed gum; AA: Acrylic acid; MBA: Methylene bisacrylamide; KPS: potassium per sulfate.

**Table 2 gels-08-00291-t002:** Dissolution data modeling showing release kinetics of captopril from hydrogels.

Formulations	Zero Order R^2^	First Order R^2^	Higuchi R^2^	Korsmeyer-Peppas	
				R^2^	*n*
CH1	0.913	0.964	0.864	0.966	0.770
CH2	0.906	0.960	0.861	0.954	0.766
CH3	0.880	0.952	0.857	0.944	0.746
CH4	0.936	0.974	0.878	0.978	0.779
CH5	0.901	0.975	0.891	0.969	0.733
CH6	0.802	0.969	0.868	0.933	0.658
CH7	0.884	0.962	0.863	0.945	0.743
CH8	0.950	0.954	0.810	0.960	0.883
CH9	0.969	0.968	0.807	0.974	0.918

**Table 3 gels-08-00291-t003:** Observations of different parameters (sign of illness, mortality, food intake, water intake and body weight) of control and hydrogel-treated group.

Parameters to Observe	Control Group (A)	Treated Group (B)	*p*-Values
Mortality	Nil	Nil	
Sign of illness	Nil	Nil	
Alertness	+	+	
Dermal toxicity/Dermal irritation	Nil	Nil	
Ocular toxicity/Eye Irritation & Lacrimation	Nil	Nil	
Light Reflex	+	+	
Food intake			
Pre-treatment	3.2 ± 1.6	3.4 ± 1.6	*p* > 0.05
Day 1	3.7 ± 1.8	2.9 ± 1.4	*p* > 0.05
Day 2	3.9 ± 2.9	3.4 ± 2.2	*p* > 0.05
Day 5	4.1 ± 1.0	3.8 ± 1.8	*p* > 0.05
Day 7	4.2 ± 1.4	4.1 ± 1.2	*p* > 0.05
Day 14	4.8 ± 1.2	4.4 ± 2.6	*p* > 0.05
Water intake			
Pre-treatment	7.8 ± 2.4	7.4 ± 2.0	*p* > 0.05
Day 1	8.4 ± 1.4	8.2 ± 1.0	*p* > 0.05
Day 2	9.2 ± 1.2	8.8 ± 1.6	*p* > 0.05
Day 5	11.2 ± 0.9	10.4 ± 1.8	*p* > 0.05
Day 7	11.6 ± 1.2	10.9 ± 2.1	*p* > 0.05
Day 14	13.2 ± 1.4	12.8 ± 1.8	*p* > 0.05
Body weight			
Pre-treatment	26.5 ± 2.4	27.5 ± 1.2	*p* > 0.05
Day 1	26.8 ± 1.2	27.9 ± 2.4	*p* > 0.05
Day 2	27.5 ± 1.6	28.3 ± 1.2	*p* > 0.05
Day 5	28.5 ± 2.2	28.8 ± 3.2	*p* > 0.05
Day 7	29.8 ± 1.4	30.4 ± 2.2	*p* > 0.05
Day 14	30.5 ± 1.6	31.2 ± 1.4	*p* > 0.05

All the values in the above-described table are expressed as mean ± SEM of eight Swiss albino mice in each group. Group A is the control; Group B is treated with (BSG-co-poly (AA)-captopril-loaded hydrogels. Nil sign indicates the absence of specified observation. Positive (+) sign indicates the presence of specified observations. *p*-values are >0.05 for all above-described results in the table, indicating that the results are statistically insignificant between both groups.

**Table 4 gels-08-00291-t004:** Biochemical plasma analysis of control and hydrogel-treated group.

Plasma Analysis	Control Group (A)	Treated Group (B)	*p*-Value
ALT (IU/L)	58 ± 2.9	55 ± 4.3	*p* > 0.05
AST (IU/L)	156 ± 8.1	144 ± 7.3	*p* > 0.05
ALP (IU/L)	123.59 ± 1.9	119.95 ± 2.4	*p* > 0.05
Cholesterol (mg/dL)	137 ± 4.3	126 ± 6.2	*p* > 0.05
Triglyceride (mg/dL)	122 ± 2.8	110 ± 4.4	*p* > 0.05
Creatinine (mg/dL)	0.43 ± 1.2	0.38 ± 0.9	*p* > 0.05
Glucose (mg/dL)	112.5 ± 0.2	116.1 ± 0.5	*p* > 0.05
Urea (mg/dL)	58 ± 3.2	62 ± 2.3	*p* > 0.05
Uric acid (mg/dL)	4.4 ± 2.6	5.6 ± 1.2	*p* > 0.05
Hemoglobin (g/dL)	13.9 ± 0.7	14.2 ± 0.5	*p* > 0.05
Hematocrit (%)	40.8 ± 3.0	42.4 ± 2.1	*p* > 0.05
RBCs (10^6^/uL)	8.4 ± 1.8	8.9 ± 1.4	*p* > 0.05
Platelets (10^3^/uL)	998 ± 2.1	992 ± 4.2	*p* > 0.05
White blood cells (10^3^/uL)	4.2 ± 0.11	4.5 ± 0.17	*p* > 0.05

All the values in the above-described table are expressed as mean ± SEM of eight Swiss albino mice in each group. Group A is the control; Group B is treated with (BSG-co-poly (AA)-captopril-loaded hydrogels. *p*-values are > 0.05 for all above-described results in the table, indicating that the results are statistically insignificant between both groups.

**Table 5 gels-08-00291-t005:** Absolute weight of different vital organs (g) in control and hydrogel-treated group.

Groups	Heart	Liver	Spleen	Kidney	Stomach
Control (A)	0.68 ± 0.03	6.25 ± 0.12	0.62 ± 0.01	0.96 ± 0.04	1.42 ± 0.24
Hydrogels treated (B)	0.57 ± 0.05	5.96 ± 0.16	0.56 ± 0.02	0.89 ± 0.02	1.36 ± 0.44

All the values in the above-described table are expressed as mean ± SEM of eight Swiss albino mice in each group. Group A is the control; Group B is treated with (BSG-co-poly (AA)-captopril-loaded hydrogels. *p*-values are > 0.05 for all above-described results in the table, indicating that the results are statistically insignificant between both groups.

## Data Availability

Not applicable.

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
