# Peer review of "Synthesis of pH-Sensitive Cross-Linked Basil Seed Gum/Acrylic Acid Hydrogels by Free Radical Copolymerization Technique for Sustained Delivery of Captopril"

_gels, 2022, doi:10.3390/gels8050291_

Round 1

Reviewer 1 Report

Dear author, please revise your manuscript to the following suggested points

I strongly recommend revising this manuscript as follows:

  1. Redraw the Chemical structure of polymer (BSG), monomer (AA), and crosslinker (MBA) in figure 1 clearly do not stretch the structure and include the hydrogel picture in figure 1.
  2. In the Introduction line number, 45 Hydrogels hold very alluring physicochemical attributes that make them 46 reasonable for a wide scope of biomedical applications cite the following recent article . Nanosystems for drug delivery of coenzyme Q10. Environmental chemistry letters, 16(1), 71-77, and Advancement of Biomaterial‐Based Postoperative Adhesion Barriers." Macromolecular bioscience 21, no. 3 (2021): 2000395.
  3. Redraw the Fig. 2. FTIR spectra of (A) BSG (B) Captopril (C) BSG-co-poly (AA) unloaded .hydrogels (D) BSG-co-poly (AA) loaded hydrogels clear neat and clean figures should be attractive, and x-axis mention clearly label the peaks what you want to demonstrate in it.
  4. In the figures 3 (i) and (ii) label the corresponding peaks and the differences you found it should be mentioned clearly in the figure.
  5. In the introduction author should discuss pH-responsive hydrogel which is used for drug delivery applications, so the following article should be cited in the introduction, very new chemistry recently reported nucleophilic substitution reaction, that showed very good pH and temperature responsiveness, the author should cite those articles in the manuscript. 1. Reactive compatibilizer mediated precise synthesis and application of stimuli-responsive polysaccharides-polycaprolactone amphiphilic co-network gels. Dually crosslinked injectable hydrogels of poly (ethylene glycol) and poly [(2-dimethylamino) ethyl methacrylate]-b-poly (N-isopropyl acrylamide) as a wound healing promoter, Self-Assembly of Partially Alkylated Dextran-graft-poly[(2-dimethylamino)ethyl methacrylate] Copolymer Facilitating Hydrophobic/Hydrophilic Drug Delivery and Improving Conetwork Hydrogel Properties. 3. Effect of Polyethylene Glycol on Properties and Drug Encapsulation–Release Performance of Biodegradable/Cytocompatible Agarose–Polyethylene Glycol–Polycaprolactone Amphiphilic Co-Network Gels and 4. Degradable/cytocompatible and pH-responsive amphiphilic conetwork gels based on agarose-graft copolymers and polycaprolactone
  6. I would like to recommend the authors perform the rheological text for your excellent pH-responsive hydrogel system that will give more clear understanding and usefulness in the different biomedical applications. 1. Frequency weep 2. G',G" vs time and flow curve.
  7. Cross-sectional SEM images should add because the given SEM images are not showing a porous structure in your pH-responsive hydrogel,
  8. The degradation study is required in this pH-responsive hydrogel system please include the data of the pH-responsive degradation profile of the hydrogel system. The author can refer to the following papers Dually crosslinked injectable hydrogels of poly (ethylene glycol) and poly [(2-dimethylamino) ethyl methacrylate]-b-poly (N-isopropyl acrylamide) as a wound healing promoter, Self-Assembly of Partially Alkylated Dextran-graft-poly[(2-dimethylamino)ethyl methacrylate] Copolymer Facilitating Hydrophobic/Hydrophilic Drug Delivery and Improving Conetwork Hydrogel Properties. 3. Effect of Polyethylene Glycol on Properties and Drug Encapsulation–Release Performance of Biodegradable/Cytocompatible Agarose–Polyethylene Glycol–Polycaprolactone Amphiphilic Co-Network Gels and 4. Degradable/cytocompatible and pH-responsive amphiphilic conetwork gels based on agarose-graft copolymers and polycaprolactone.
  9. I would like to strongly recommend to the authors please improve the representation of the data, this article has excellent data but the representation seems not perfect please make the manuscript attractive.

Reviewer 2 Report

Dear author,

Here are my comments for your paper:

  1. Discuss the novelty and added value of the paper with respect to existent literature.
  2. Discuss the copolymerization mechanism.
  3. Base line in FTIR is not very accurate. Add also the important peaks on the plots.
  4. Why is important the thermal analysis taking into account the target application?
  5. XRD: "In captopril-loaded hydrogel, the reduction in intensity and
    sharpness of peaks indicating that drug was entrapped in the formulated hydrogel polymeric network". This is speculative and not proven. XRD data must be processed in a specific software (Origin for example) to smoothen the spectra and have high-quality diffractograms. The Ox and Oy axis are not clearly seen, this must be improved.
  6. Did you try to obtain porous hydrogels by freeze-drying technique? This could be interesting for drug uptake. The induced porosity by freeze-drying is higher and more proper for the target drug.
  7. What about the rheological properties of the hydrogels? Elastic and storage moduli...
  8. Did you try to upload the drug directly in the copolymerization process?
  9. Discuss the porosity degree by BET analysis or similar techniques.

Round 2

Reviewer 1 Report

It seems Authors has revised all suggested points 

Reviewer 2 Report

Dear authors,

The paper has now been improved and could be accepted.